# Antiproliferative and Antiangiogenic Properties of New VEGFR-2-targeting 2-thioxobenzo[*g*]quinazoline Derivatives (In Vitro)

**DOI:** 10.3390/molecules25245944

**Published:** 2020-12-15

**Authors:** Hatem A. Abuelizz, Mohamed Marzouk, Ahmed H. Bakheit, Hanem M. Awad, Maha M. Soltan, Ahmed M. Naglah, Rashad Al-Salahi

**Affiliations:** 1Department of Pharmaceutical Chemistry, College of Pharmacy, King Saud University, P.O. Box 2457, Riyadh 11451, Saudi Arabia; Habuelizz@ksu.edu.sa (H.A.A.); abujazz76@gmail.com (A.H.B.); 2Department of Tanning Materials and Leather Technology, National Research Centre, 33 El-Bohouth St. (Former El-Tahrir St.), Dokki, Cairo 12622, Egypt; msmarzouk@yahoo.co.uk (M.M.); Hanem_awad@yahoo.com (H.M.A.); 3Department of Chemistry, Faculty of Science and Technology, El-Neelain University, P.O. Box 12702, Khartoum 11121, Sudan; 4Biology Unit, Central Laboratory for Pharmaceutical and Drug Industries Research Division, Chemistry of Medicinal Plants Department, National Research Centre, El Bohouth St. 33, Dokki, Cairo 12622, Egypt; mahasoltan@netscape.net; 5Department of Pharmaceutical Chemistry, Drug Exploration and Development Chair (DEDC), College of Pharmacy, King Saud University, Riyadh 11451, Saudi Arabia; amnaglah@gmail.com; 6Peptide Chemistry Department, Chemical Industries Research Division, National Research Centre, 33 El-Bohouth St. (Former El-Tahrir St.), Dokki, Cairo 12622, Egypt

**Keywords:** benzoquinazolines, MCF-7, HepG2, MTT assay, apoptosis, VEGFR-2, QSAR, docking study

## Abstract

A series of 3-ethyl(methyl)-2-thioxo-2,3-dihydrobenzo[*g*]quinazolines (**1**–**17**) were synthesized, characterized, and evaluated in vitro for their antiangiogenesis VEGFR-2-targeting, antiproliferative, and antiapoptotic activities against breast MCF-7 and liver HepG2 cells. Flow cytometry was used to determine cancer-cell cycle distributions, and apoptosis was detected using annexin-V-FITC (V) and propidium iodide (PI) dyes. Fluorescence microscopy, in combination with Hoechst staining was used to detect DNA fragmentation. Most of the tested benzo[*g*]quinazolines demonstrated promising activity (IC_50_ = 8.8 ± 0.5–10.9 ± 0.9 μM) and (IC_50_ = 26.0 ± 2.5–40.4 ± 4.1 μM) against MCF-7 and HepG2, respectively. Doxorubicin was used as a reference drug. Compounds **13**–**15** showed the highest activity against both cancer cell lines. Differential effects were detected by cell-cycle analysis, indicating similarities in the actions of **13** and **14** against both MCF7 and HepG2, involving the targeting of G1 and S phases, respectively. Compound **15** showed similar indices against both cells, indicating that its cytotoxicity toward the examined cancer cells could be unselective. Interestingly, **14** and **15** showed the highest apoptosis (30.76% and 25.30%, respectively) against MCF-7. The DNA fragmentation results agreed well with the apoptosis detected by flow cytometry. In terms of antiangiogenesis activity, as derived from VEGFR-2 inhibition, **13** and **15** were comparable to sorafenib and effected 1.5- and 1.4-fold inhibition relative to the standard sorafenib. A docking study was conducted to investigate the interaction between the synthesized benzo[*g*]quinazolines and the ATP-binding site within the catalytic domain of VEGFR-2.

## 1. Introduction

Angiogenesis plays a substantial role in cancer development and relies on the appearance of specific mediators that start a cascade of events resulting in microvessel formation [1]. Vascular endothelial growth factor receptor-2 (VEGFR-2) is one of the main targets within the angiogenesis-related kinases and is the main endothelial species required for tumor neovascularization [2,3,4]. Such kinases exert their biological effects by protecting against apoptosis-inducing conditions, increasing vessel permeability, and promoting endothelial cell proliferation enhancement, migration, and differentiation. Fighting cancer angiogenesis is one of the main focuses of effective cancer remedy. Thus, obstruction of erratic angiogenesis by specific inhibitors that target receptor tyrosine kinases is of pronounced interest in medicinal chemistry research. By specifically preventing angiogenesis by VEGFR-2 inhibitors, Figure 1A represents an attractive strategy for the treatment of VEGFR-2-mediated tumor growth, as well as addressing drug resistance [2,3,4].

VEGFR-2 inhibitors are classified into three types (I, II, and III). Type-I are ATP-competitive inhibitors such as sunitinib, which bind to the region that accommodates ATP’s adenine ring [5]. Sorafenib belongs to the second type of inhibitors (II), which cannot bind at the adenine binding site but instead bind adjacent to the hydrophobic pocket [6]. Type-III inhibitors, such as vatalanib, are covalent inhibitors that bind to cysteine amino acid residues and prevent binding of ATP at the binding site [7,8]. Blocking the receptor dimerization and autophosphorylation processes provides a basis for discovering novel VEGFR-2 inhibitors. For VEGFR-2, the catalytic domain is the ATP site for binding. It is a bi-lobed structure with a small N-lobe and a large C-lobe. These lobes are linked via a linker that consists of a hinge region and a convex motif region. The binding of ATP is dependent on the confirmation of these lobes (Figure 1C) [9,10].

Molecules featuring the benzo[*g*]quinazoline scaffold are widely studied in medicinal chemistry because of their numerous biological and chemical properties [11,12,13,14,15,16,17,18,19,20]. In our studies on benzo[*g*]quinazoline chemistry, many derivatives have been identified as potent α-glucosidase inhibitors with promising antidiabetic effects [17] and antimicrobial agents that show excellent activities against both Gram-positive and -negative bacteria along with strong antifungal activities [11,14,15]. Furthermore, some derivatives of benzo[*g*]quinazolines show high antiviral activity against herpes simplex (HSV-1 and 2), coxsackie (CVB4), and HAV viruses [12,19]. Benzo[*h*]quinazolines have been reported as potent tyrosine kinase inhibitors that exhibit significant cytotoxicity against human carcinoma HT29, HCT116, and A549 cell lines [21], whereas benzo[*f*]quinazoline derivatives have been demonstrated to be potent thymidylate synthase inhibitors [22]. Moreover, benzo[*g*]quinazolines bearing sulfonamide, alkoxy, amino, and thioxo functional groups have shown high affinities toward the anaplastic lymphoma kinase receptor and demonstrate potent effects against EGFR and HER2 cells [16].

In our previous studies, we have evaluated some 2-thioxobenzo[*g*]quinazoline derivatives for their cytotoxicity activities against several carcinoma cell lines, i.e., A549, PC-3, HCT-116, Hep-G2, and MCF-7 [13]. These results encouraged us to consider the benzo[*g*]quinazoline core as a template for the design of further derivatives with potent cytotoxic activities. Moreover, chemical modification of 2-thioxobenzo[*g*]quinazoline structures provided us with valuable insight into the properties required for potent benzo[g]quinazoline-based anticancer agents. VEGF/VEGFR-2 binding is an early event in the angiogenic cascade, so targeting VEGFR-2 receptors and/or its ligand VEGF are considered good strategies for arresting cancer metastasis. We continued our attention in drug discovery, regarding the antiangiogenic therapy, besides our attention to exploring more anticancer agents. 

Herein, 3-ethyl(methyl)-2-thioxobenzo[*g*]quinazolines (**1**–**17**) were synthesized and evaluated for their cytotoxicities against human breast MCF-7 and liver HepG2 carcinoma cells and their abilities to trigger apoptosis and restrict cancer-cell migration. Furthermore, the most active benzoquinazolines were evaluated as antiangiogenic chemotherapeutic agents targeting the VEGFR-2 tyrosine kinase receptor (Figure 1B). Finally, a molecular docking study was conducted to rationalize the structure-activity relationship (SAR) of the synthesized benzo[*g*]quinazolines and predict their interactions with the ATP-binding site in the catalytic domain of VEGFR-2. The QSAR was carried out to explore the relationships between the benzo[*g*]quinazolines molecular structures and their cytotoxicity findings by developing prediction models among the known anti HepG2 and MCF-7 agents then to predict the cytotoxicity of the benzo[*g*]quinazolines.

## 2. Results and Discussion

### 2.1. Chemistry

Using previously reported synthetic procedures for preparation of their analogues [13,17,23], the current 3-ethyl(methyl)-2-thioxobenzo[*g*]quinazolin-4(3*H*)-ones (**1**–**17**) were obtained in good yields (see Section 3.1.1). The structures of all products were established using HREI-MS and NMR analyses. The benzoquinazoline core was confirmed by observation of the three aromatic resonance pairs in the range of 8.8–7.5 ppm. The first is two singlets for H-5 and H-10; the second is for H-6 and H-9 in the form of two broad doublets, and the third is two identical broad triplets at around 7.60 and 7.50 ppm for H-7 and H-8, respectively [13,17,23,24]. The core structure was confirmed from the 12 typical ^13^C-resonances, including the most downfield ones at around 161.0 and 156.0 ppm, assignable for C-4 and C-2, respectively. In the case of the 2-thioxo derivatives (**1**, **2**), C-2 was interpreted at 175.0–176.0 ppm. Further confirmation was achieved according to the intrinsic NMR data corresponding to the *S*-alkyl/benzyl or *N*-alkyl (Me or Et) substituents located at positions 2 and 3, respectively. Concerning the 2-*S*-benzyl derivatives (**4**–**8**, **10**–**13**), the CH_2_-group presented singlets at 4.5–4.65 and 35.0–36.0 ppm in the ^1^H and ^13^C NMR spectra, respectively. The 3-*N*-ethyl products (**3**, **9**) were unambiguously identified from the characteristic A2 × 3 quartet and triplet around 4.10 and 1.27 ppm with ^13^C-resonances around 41.0 and 13.7 ppm, respectively, while 2-*S*-ethyl resonances were observed at 3.3 and 1.4 in the ^1^H NMR and at 26.5 and 14.5 ppm in ^13^C NMR spectra. The 3-*N*-Me group was assigned to the singlets around 3.5 ppm and 30.5 ppm in the ^1^H and ^13^C NMR spectra, respectively (**1**–**8**, **16**). The 2-hydrazinyl products (**16**, **17**) were confirmed through two characteristic singlets at 9.55 and 6.22 ppm for –NH-NH_2_, whereas the aryl C-CH_3_ group was assigned at about 2.3 and 21.4 in the ^1^H and ^13^C NMR, respectively (**12**, **5**). Similarly, the confirmation of **14** and **15** based on the characteristic resonances of the *N*-ethylpiperidine, propyl and isoindoline moieties (see experimental data). 

### 2.2. Biology 

#### 2.2.1. Antiproliferative Activity

Selected compounds were evaluated in vitro for their antiproliferative activities against HepG2 and MCF-7 human cancer-cell lines using MTT assays. Cell viabilities and IC_50_ values were obtained and compared with those of the control, doxorubicin (Table 1).

All the compounds presented dose-dependent antiproliferative activities against both cell varieties (Appendix A). The IC_50_ values for compounds **1**–**17** are summarized in Table 1. For HepG2 cancer cells, benzoquinazolines **13**, **14**, and **15** exhibited potent activities (IC_50_ = 27.5 ± 2.1, 27.7 ± 2.5 and 26.0 ± 2.5 μM, respectively); compounds **6** and **16** had comparable activities (IC_50_ = 28.8 ± 2.6 and 28.9 ± 2.3 μM); **3**–**5**, **8**, **9**, **11**, **12**, and **17** exhibited slightly lower activities; and compounds **1**, **2**, **7**, and **10**, showed moderate activities relative to that of the positive control (IC_50_ = 28.5 ± 1.9 μM). Regarding the breast cancer cells (MCF-7); benzoquinazolines **3**, **4**, **7**, **9**, **10**, **13**–**15**, and **17** were more potent (IC_50_ = 9.6 ± 0.5–10.2 ± 1.1 μM) than doxorubibcin (IC_50_ = 10.3 ± 0.8). The rest of the compounds showed slightly lower activities relative to that of the positive control (Table 1). Thus, our results demonstrated that compounds **13**, **14**, and **15** were the most active against the two examined cell lines.

The results in Table 1 indicate that the chemical changes to the lead structures **1** and **2** have considerable impacts on their cytotoxicity profiles for all the derivatives (**3**–**17**). Particularly, the S-alkylated products (**3**–**5**, **7**–**15**) showed increasing cytotoxicities against MCF-7, and improved activity profiles against HepG2. Furthermore, hydrazinolysis of **1** and **2** into **16** and **17** resulted in remarkable changes in activity. Variations in the position of substitution on the benzyl ring (**5**–**8**, **11**–**13**) also affected cytotoxicity, where the presence of the chlorine atom in **6** resulted in a significant increasing in cytotoxicity against HepG2 cells in comparison with the corresponding cyano and methoxy derivatives (**7** and **8**). Furthermore, the heteroalkyl moiety provided a significant enhancement in cytotoxicity, as indicated by compounds **14** and **15**. Thus, the two cell lines studied appeared to be sensitive toward the antiproliferative properties of most of the investigated benzoquinazolines, with **13**, **14**, and **15** exhibiting the most noticeable effects

#### 2.2.2. Flow Cytometry Cell-Cycle Analysis

Based on cytotoxicity screening (Table 1), we select compounds **13**–**15** for further investigation of anticancer activity against MCF-7 and HepG2 cancer cells. In addition, **10** was tested against MCF-7 cells. Analyzing cells by flow cytometry allows the detection of their different effects on cell-cycle distribution. Table 2 showed the percentage cell accumulation for the G0/G1, S, and G2/M phases at their IC_50_ concentrations, as well as apoptosis percentages for the pre-G1 phase (Figure 2A,B and Figure 3). The results revealed the capability of compound **10** to arrest MCF-7 cells at the G2/M phase (34.6% compared to 9.5% for the vehicle control). Compounds **13** and **14** exerted their effects similarly, with MCF-7 cell accumulation at the G1 phase of approximately 62.4% and 66.2%, respectively (Figure 4). Both were remarkably capable of elevating the percentages of HepG2 cells at the S phase (55.3% and 52.3%, respectively). The selectivities of **13** and **14** for the G1 phase in MCF-7 cells were highlighted in the present study, while both trigger some cell-cycle arrest at the S phase. Conversely, the cell-cycle-arrest of compound **15** did not show phase selectivity. Treating MCF-7 and HepG2 cells with **15** resulted in accumulations of 28.0% and 39.2%, respectively, indicating non-selectivity.

#### 2.2.3. Detection of Apoptosis

##### Double Staining with V and PI Dyes

To investigate the type of cell death induced, the treated MCF-7 and HepG2 cells were stained with V and PI. The apoptotic cells are indicated by V+/PI− and V+/PI+ in Figure 2 and Figure 3. Interestingly, apoptosis was evidenced by all the tested compounds (**10, 13**–**15**) against both MCF-7 and HepG2 cells. Table 2 reveals 21.67–30.76% apoptosis for MCF-7 cells, with the best induction being observed for **14** (Figure 4A). Induction of apoptosis in the HepG2 cells ranges from 16.39 to 21.38%, with **13** being the best inducer (Figure 4B). However, it should be denoted that, the percentages of the necrotic MCF-7 cells recorded increment upon the separate treatment, with the IC_50_ of the selected compounds (about 8–15%), while **13** was the worst. Conversely, their equivalents from the treated HepG2 showed about 6%.

##### Hoechst 33258 Nuclear Staining-DNA Fragmentation

The nuclear morphological changes were detected by fluorescence microscopy upon staining the treated cells with Hoechst 33258 dye. DMSO, as a vehicle, introduces a uniform light-blue staining and intact cell membranes (Figure 5). Conversely, cells treated for 48 h at the IC_50_ of each compound were reduced in size. Bright-blue fluorescence was observed due to chromatin condensation, except in the case of MCF-7 treated with compound **13**, where elevation in the necrotic cells by 15% was observed (as shown by flow cytometry). Their morphology, size, and blue color did not differ more than those of the MCF-7 vehicle. Interestingly, **15** caused similar changes for both cells. 

##### In Vitro Inhibition of VEGFR-2

Compounds **10** and **13**–**15** were further evaluated for their antiangiogenic activities. Figure 6 and Table 3 show the determined IC_50_ values (46.6 ± 2.8 and 44.4 ± 2.6 nM, respectively)) for **13** and **15**, which are comparable to that of sorafenib (31.1 ± 1.8 nM).

Our results confirmed the widely accepted fact that VEGF prevents apoptosis [25]. In the present study, inhibition of the VEGFR-2 enzyme and induction of apoptosis for both tested cell lines were observed for compounds **10** and **13**–**15**. The VEGFR-2 inhibition activity relative to sorafenib increased in the order of **15** (1.4-fold) > **13** (1.5-fold) > **14** (2.0-fold) > **10** (4.5-fold). The induction of apoptosis for MCF-7 (Table 2) follows the order **14** (30.76%) > **15** (25.3%) > **13** (22.22%) > **10** (21.67%). Conversely, HepG2 apoptosis followed the order **13** (21.38%) > **15** (17.84%) > **14** (16.39%).

### 2.3. Molecular Docking

The molecular design of the benzo[*g*]quinazolines as VEGFR-2 inhibitors was based on the binding mode of sorafenib with VEGFR-2, which was obtained from the crystallographic complex available in the Protein Data Bank (4ASE.pdb, https://www.rcsb.org/structure/4ASE). The docking study was performed to investigate the interaction between the synthesized benzo[*g*]quinazolines and the ATP-binding site, which is located between the *N*-terminal and the *C*-terminal lobes within the catalytic domain of VEGFR-2 (Figure 7A,B). Many kinase inhibitors act as ATP mimetics and compete with cellular ATP for binding sites, thus suppressing autophosphorylation [9,10]. The VEGFR-2 ATP-binding pocket residues, as shown in Figure 7B, were HYD I (encapsulated by Leu-840, Phe-918, and Gly-922), the HYD II (encapsulated by Leu-889, Ile-892, Val-8 98, and Ile-1044), and the linker (encapsulated by Ala-866, Val-914, Leu-1035, and Cys-1045). Thus, ATP-binding site of VEGFR-2 was mainly established by hydrophobic residues in HYD I and HYD II (Figure 7 and Figure 8A). 

The docking protocol was confirmed by re-docking of the co-crystallized ligand in the vicinity of the active site of the enzyme with binding affinity = −11.0317 kcal/mol and RMSD = 0.34016 (Table 4, Figure 7B). The docking results revealed that **15** and **13** were the most potent inhibitors (Figure 7A). Compound **15** demonstrated a good Z-score in QSAR (Appendix A) and a high binding affinity of −9.669 kcal/mol. Moreover, **15** tightly bound to the key amino acids Asp1046, Phe1047, and Cys919 through the formation of H bonding and Leu840, Val848, and Phe1047 via the formation of π-π interactions, as shown in Figure 8B. 

Compound **13** interacted via H bonds with Asp1046 and CYS1045 and with VAL848 via π-π interactions in the active site of VEGFR-2 (Figure 9A) with high binding affinity −8.133 kcal/mol (Table 4). Compound **14** bound with Phe918 and Cys919 via H bonds and with Leu 840 and Val 848 through π-π interactions with the high binding affinity −8.307 kcal/mol (Figure 9B). Moreover, the configuration results for compounds **13** and **15** revealed that both compounds were type-II inhibitors due to their binding in the DFG region adjacent to the HYD II region and called the allosteric site. The other compounds were identified as type-I inhibitors, as their interactions were mainly with the hinge-region residues Cys919 and Leu840 adjacent to HYD I, as shown in Table 4. 

### 2.4. QSAR Study

Partial least squares (PLS) were employed to analyze the chemical space of the training set. After PLS with the 364 Molecular operating environment (MOE) descriptors for 71 training set modeling compounds, the first three most important components (PEOE_VSA+0, PEOE_VSA-1, and SMR_VSA1) and (PEOE_VSA+0, PEOE_VSA+1, and SMR_VSA-0) for HepG2 and MCF-7, respectively, were chosen to generate a 3D plot (Figure 10A,B) for these compounds. The compound activities were distinguished well, and were labeled in different colors according to their biological response. Therefore, we retained these three descriptors and combined them with two other descriptors to build QSAR models (Appendix A).

Before building the QSAR models, all descriptor’s correlations must be considered. If one descriptor is collinear with one or more descriptors, i.e., the correlation result is higher than 0.5, these descriptors should not be considered together. The selected descriptors had little mutual correlation and can be used in the same equation (Appendix A). PEOE_VSA+5 and PEOE_VSA-1 exemplified the sum of van der Waals surface areas, where the partial charges are in the range (0.25, 0.30) and (−0.10, −0.05), respectively. SMR_VSA1, SMR_VSA4, and SMR_VSA7 were molar refractivity descriptors and had a high correlation with molecular polarizability [26]. They were calculated in the ranges (0.11–0.26), (0.39–0.44), and >0.56, respectively. The correlation coefficients for HepG2 and MCF-7 and their data had acceptable R^2^, XR^2^, RMSE, and the cross-validated RMSE values for all compounds, as shown by Equations (5) and (6) (Appendix A). A high value of q2 (for instance, q2 > 0.5), as an indicator or even as the ultimate proof that model, is considered highly predictive by many authors. The Equations (3) and (4) models (Appendix A) have the following statistical results: $Q_{\mathrm{CV}}^{2}$ = 0.62709, XRMSE = 0.21027, $Q_{\mathrm{CV}}^{2}$ = 0.7277, and XRMSE = 0.2822, which suggests that these models are highly predictive and reliable for the prediction of biological activity. The pIC_50_ data indicate that the range is wide, from 4.2 to 0.56 and 4.97 to 5.02 for HepG2 and MCF-7, respectively (Appendix A). The structures of the target benzoquinazolines were diverse, with different substitutions. Thus, the width of the data and the variety of the structures can help to increase the prediction ability and universality of the QSAR model. Figure 10C,D shows that the curve for the experimental pIC_50_ is similar to that of the calculated pIC50, and the deviations mainly fluctuate from −0.5 to 0.5 for both HepG2 and MCF-7. Approximately 80% of the deviations are around 0 for both, indicating that the QSAR model had high quality and stability from the training set.

The predicted pIC50 values for the training and test sets were plotted against the experimental pIC50 values (Figure 11 and Figure 12). The predicted pIC_50_ results obtained for both the modeling set were in good agreement with the experimental results obtained for biological activity against HepG2 and MCF-7. The residual values obtained between predicted and experimental pIC_50_ values were very low. The coefficient of determination (*R*^2^) was close to 1.0, indicating that the model represents a very high percentage of the response variable (descriptor) variation, certainly high enough for a robust QSAR model. The obtained values (0.71947 and 0.85989) indicated that 71.947 and 85.989% of the variation for HepG2 and MCF-7, respectively, resided in the residual, indicating that the model is very good.

## 3. Materials and Methods 

### 3.1. Chemistry

#### 3.1.1. Synthesis of **1**–**17**

The general procedures for the preparation of compounds **1**–**17** were previously reported for the preparation of their analogues [13,17]. Briefly, in the basic medium, the reaction of ethyl(methyl)isothiocyanate with 3-amino-2-naphthaoic acid under a refluxing condition in DMF for 3–5 h afforded the parent intermediates **1** and **2** in good yields (Scheme 1). When compound **1** or **2** was treated with the appropriate alkyl/heteroalkyl halide in the presence of a base at 80 °C for 20 h, benzo[*g*]quinazolines **3**–**15** were obtained in good to high yields. Hydrazinolysis of **1** and **2** in boiling DMF for 15–18 h afforded **16** and **17**, respectively. Their characterization data are reported herein as follows:

##### 3-Methyl-2-thioxo-2,3-dihydrobenzo[*g*]quinazolin-4(1*H*)-one (**1**) 

White amorphous powder (86%); mp > 300 °C; ^1^H NMR (500 MHz, DMSO-*d_6_*): *δ* 13.02 (br s, 1H, -NH), 8.71 (s, 1H, H-5), 8.14 (br s, 1H, H-6), 7.95 (br s, 1H, H-9), 7.77 (s, 1H, H-10), 7.64 (br s, 1H, H-7), 7.51 (br s, 1H, H-8), 3.72 (s, 3H, N-CH_3_); ^13^C NMR (125 MHz, DMSO-*d_6_*): *δ* 175.9 (C-2), 160.2 (C-4), 136.6 (C-4b,9a), 135.3 (C-5a), 130.5 (C-5), 129.9 (C-6), 129.5 (C-8), 127.5 (C-4a), 125.9 (C-9), 115.8 (C-7), 111.3 (C-10), 33.5 (N-CH_3_); HRMS-MS: *m*/*z* calcd. for C_13_H_10_N_2_OS (M)^•+^ 242.0514, found 242.0531 [23,24].

##### 3-Ethyl-2-thioxo-2,3-dihydrobenzo[g]quinazolin-4(1*H*)-one (**2**) 

White amorphous powder (82%); mp 296 °C; ^1^H NMR (700 MHz, DMSO-*d_6_*): *δ* 12.96 (s, 1H, -NH-), 8.69 (s, 1H, H-5), 8.12 (br d, *J* = 8 Hz, 1H, H-6), 7.93 (br d, *J* = 8 Hz, 1H, H-9), 7.74 (s, 1H, H-10), 7.62 (br t, *J* = 7.4 Hz, 1H, H-7), 7.50 (br t, *J* = 7.4 Hz, 1H, H-8), 4.49 (t, q = 7 Hz, 2H, H-1′), 1.27 (t, *J* = 7 Hz, 3H, H-2′); ^13^C NMR (175 MHz, DMSO-*d_6_*): *δ* 175.4 (C-2), 159.5 (C-4), 136.6 (C-4b,9a), 135.3 (C-5a), 130.0 (C-5), 129.9 (C-6), 129.7 (C-8), 127.7 (C-4a), 126.1 (C-9), 116.0 (C-7), 111.5 (C-10), 41.4 (C-1′), 12.7 (C-2′); HRMS-MS: *m*/*z* calcd. for C_14_H_12_N_2_OS (M)^•+^ 256.0670, found 256.0693 [23,24].

##### 2-(Ethylthio)-3-methylbenzo[*g*]quinazolin-4(3*H*)-one (**3**) 

White amorphous powder (75%); mp 115–117 °C; ^1^H NMR (500 MHz, DMSO-*d6*): *δ* 8.76 (s, 1H, H-5), 8.17 (br d, *J* = 8.3 Hz, 1H, H-6), 8.05 (s, 1H, H-10), 8.04 (br d, *J* = 8.3 Hz, 1H, H-9), 7.63 (br t, *J* = 7.3 Hz, 1H, H-7), 7.53 (br t, *J* = 7.3 Hz, 1H, H-8), 3.45 (s, 3H, N-CH_3_), 3.28 (q, *J* = 7.3 Hz, 2H, H-1′), 1.41 (t, *J* = 7.3 Hz, 3H, H-2′); ^13^C NMR (125 MHz, DMSO-*d6*): *δ* 161.5 (C-4), 156.5 (C-2), 142.8 (C-4b), 136.8 (C-5a), 130.8 (C-9a), 129.7 (C-5), 128.9 (C-6), 128.3 (C-8), 128.1 (C-4a), 126.3 (C-9), 123.1 (C-7), 118.8 (C-10), 30.4 (N-CH_3_), 26.4 (C-1′), 14.5 (C-2′); HRMS-MS: *m*/*z* calcd. for C_15_H_14_N_2_OS (M)^•+^ 270.0827, found 270.0872 [24].

##### 2-(Benzylthio)-3-methylbenzo[*g*]quinazolin-4(3*H*)-one (**4**)

Pale yellow amorphous powder (71%); mp 150–152 °C; ^1^H NMR (700 MHz, DMSO-*d_6_*): *δ* 8.81 (s, 1H, H-5), 8.19 (br s, 2H, H-6, 10), 8.11 (br s, 1H, H-9), 7.67 (br s, 1H, H-7), 7.58 (br s, 3H, H-8, 2′/6′), 7.36 (br s, 2H, H-3′/5′), 7.28 (br s, 1H, H-4′), 4.61 (s, 2H, -CH_2_-7′), 3.52 (s, 3H, N-CH_3_); ^13^C NMR (175 MHz, DMSO-*d_6_*): *δ* 161.5 (C-4), 156.2 (C-2), 142.7 (C-4b), 137.3 (C-1′), 136.8 (C-9a), 130.9 (C-5a), 130.1 (C-5), 129.9 (C-3′/5′), 129.8 (C-6), 129.1 (C-8), 128.9 (C-2′/6′), 128.4 (C-4a), 128.1 (C-4′), 126.4 (C-9), 123.2 (C-7), 118.8 (C-10), 35.9 (-CH_2_-7′), 30.4 (N-CH_3_); HRMS-MS: *m*/*z* calcd. for C_20_H_16_N_2_OS (M)^•+^ 332.0983, found 332.0991.

##### 3-Methyl-2-((3-methylbenzyl)thio)benzo[*g*]quinazolin-4(3*H*)-one (**5**)

White amorphous powder (70%); mp 155–157 °C; ^1^H NMR (500 MHz, DMSO-*d6*): *δ* 8.81 (s, 1H, H-5), 8.20 (br d, *J* = 8.4 Hz, 1H, H-6), 8.18 (s, 1H, H-10), 8.10 (br d, *J* = 8.4 Hz, 1H, H-9), 7.66 (br t, *J* = 7.6 Hz, 1H, H-7), 7.56 (br t, *J* = 7.6 Hz, 1H, H-8), 7.38 (br s, 1H, H-2′), 7.36 (br d, *J* = 7.5 Hz, 1H, H-6′), 7.24 (br t, *J* = 7.5 Hz, 1H, H-5′), 7.08 (br d, *J* = 7.5 Hz, 1H, H-4′), 4.57 (s, 2H, -CH_2_-7′), 3.52 (s, 3H, N-CH_3_), 2.30 (s, 3H, Ar-*Me*); ^13^C NMR (125 MHz, DMSO-*d6*): *δ* 161.5 (C-4), 156.2 (C-2), 142.7 (C-4b), 138.1 (C-1′), 136.9 (C-5a), 136.8 (C-3′), 130.9 (C-9a), 130.5 (C-5), 129.8 (C-6), 129.0 (C-2′), 128.9 (C-8), 128.6 (C-5′), 128.4 (C-4a), 128.1 (C-4′), 126.9 (C-9), 126.4 (C-6′), 123.2 (C-7), 118.9 (C-10), 36.0 (-CH_2_-7′), 30.4 (N-CH_3_), 21.4 (Ar-*Me*); HRMS-MS: *m*/*z* calcd for C_21_H_18_N_2_OS (M)^•+^ 346.1140, found 346.1155.

##### 2-((4-Chlorobenzyl)thio)-3-methylbenzo[*g*]quinazolin-4(3*H*)-one (**6**)

Pale yellow amorphous powder (78%); mp 190–192 °C; ^1^H NMR (500 MHz, DMSO-*d6*): *δ* 8.81 (s, 1H, H-5), 8.20 (br d, *J* = 8.3 Hz, 1H, H-6), 8.19 (s, 1H, H-10), 8.10 (br d, *J* = 8.3 Hz, 1H, H-9), 7.65 (br t, *J* = 7.3 Hz, 1H, H-7), 7.59 (d, *J* = 8.4 Hz, 2H, H-3′/5′), 7.56 (br t, *J* = 7.3 Hz, 1H, H-8), 7.41 (d, *J* = 8.4 Hz, 2H, H-2′/6′), 4.60 (s, 2H, -CH2-7′), 3.52 (s, 3H, N-CH_3_); ^13^C NMR (125 MHz, DMSO-*d6*): *δ* 161.5 (C-4), 155.9 (C-2), 142.6 (C-4b), 136.8 (C-5a), 132.4 (C-1′), 131.7 (C-3′/5′), 131.2 (C-4′), 130.9 (C-9a), 129.8 (C-5), 129.1 (C-6), 128.8 (C-2′/6′), 128.4 (C-8), 128.1 (C-4a), 126.4 (C-9), 123.2 (C-7), 118.7 (C-10), 35.0 (-CH2-7′), 30.4 (N-CH_3_); HRMS-MS: *m*/*z* calcd. for C_20_H_15_ClN_2_OS (M)^•+^ 366.0594, found 366.0607.

##### 4-(((3-Methyl-4-oxo-3,4-dihydrobenzo[*g*]quinazolin-2-yl)thio)methyl)benzonitrile (**7**)

White amorphous powder (75%); mp 201–203 °C; ^1^H NMR (500 MHz, DMSO-*d_6_*): *δ* 8.79 (s, 1H, H-5), 8.19 (br d, *J* = 7.5 Hz, 1H, H-6), 8.18 (s, 1H, H-10), 8.07 (br d, *J* = 7.5 Hz, 1H, H-9), 7.94 (d, *J* = 8.5 Hz, 2H, H-3′/5′), 7.73 (d, *J* = 8.5 Hz, 2H, H-2′/6′), 7.65 (br t, *J* = 7.5 Hz, 1H, H-7), 7.55 (br t, *J* = 7.5 Hz, 1H, H-8), 4.64 (s, 2H, -CH2-7′), 3.51 (s, 3H, N-CH_3_); ^13^C NMR (125 MHz, DMSO-*d_6_*): *δ* 161.5 (C-4), 155.6 (C-2), 143.8 (C-4b), 141.4 (C-1′), 136.8 (C-9a), 132.7 (C-3′/5′), 131.0 (C-5a), 130.7 (C-2′/6′), 129.7 (C-5), 128.8 (C-6), 128.4 (C-8), 128.1 (C-4a), 126.4 (C-9), 123.2 (C-7), 119.0 (C≡N), 118.8 (C-10), 110.9 (C-4′), 35.5 (-CH2-7′), 30.4 (N-CH_3_),; HRMS-MS: *m*/*z* calcd. for C_21_H_15_N_3_OS (M)^•+^ 357.0936, found 357.0950. 

##### 2-((3-Methoxybenzyl)thio)-3-methylbenzo[*g*]quinazolin-4(3*H*)-one (**8**)

White amorphous powder (75%); mp 188–190 °C; ^1^H NMR (500 MHz, DMSO-*d6*): *δ* 8.82 (s, 1H, H-5), 8.21 (br d, *J* = 8.7 Hz, 1H, H-6), 8.20 (s, 1H, H-10), 8.11 (br d, *J* = 8.7 Hz, 1H, H-9), 7.67 (br t, *J* = 7.3 Hz, 1H, H-7), 7.57 (br t, *J* = 7.3 Hz, 1H, H-8), 7.27 (br t, *J* = 8.1 Hz, 1H, H-5′), 7.17 (br s, 1H, H-2′), 7.14 (br d, *J* = 8.1 Hz, 1H, H-6′), 6.86 (br d, *J* = 8.3 Hz, 1H, H-4′), 4.59 (s, 2H, -CH2-7′), 3.75 (s, 3H, -OCH_3_), 3.53 (s, 3H, N-CH_3_); ^13^C NMR (125 MHz, DMSO-*d6*): *δ* 161.5 (C-4), 159.7 (C-3′), 156.2 (C-2), 142.7 (C-4b), 138.8 (C-1′), 136.8 (C-5a), 130.9 (C-9a), 130.1 (C-5′), 129.8 (C-5), 129.1 (C-6), 128.4 (C-8), 128.1 (C-4a), 126.4 (C-9), 123.2 (C-7), 122.1 (C-6′), 118.8 (C-10), 115.5 (C-4′), 113.4 (C-2′), 55.5 (-OCH3), 35.9 (-CH2-7′), 30.4 (N-CH_3_); HRMS-MS: *m*/*z* calcd. for C_21_H_18_N_2_O_2_S (M)^•+^ 362.1089, found 362.1104.

##### 3-Ethyl-2-(ethylthio)benzo[*g*]quinazolin-4(3*H*)-one (**9**) 

White amorphous powder (73%); mp 107–109 °C; ^1^H NMR (500 MHz, DMSO-*d6*): *δ* 8.79 (s, 1H, H-5), 8.18 (br d, *J* = 8.3 Hz, 1H, H-6), 8.10 (s, 1H, H-10), 8.08 (br d, *J* = 8.3 Hz, 1H, H-9), 7.65 (br t, *J* = 7.8 Hz, 1H, H-7), 7.55 (br t, *J* = 7.8 Hz, 1H, H-8), 4.10 (q, *J* = 7.1 Hz, 2H, H-1″), 3.31 (q, *J* = 7.3 Hz, 2H, H-1′), 1.43 (t, *J* = 7.3 Hz, 3H, H-2′), 1.31 (t, *J* = 7.1 Hz, 3H, H-2″); ^13^C NMR (125 MHz, DMSO-*d6*): *δ* 161.1 (C-4), 155.7 (C-2), 142.8 (C-4b), 136.8 (C-5a), 130.9 (C-9a), 129.7 (C-5), 129.0 (C-6), 128.3 (C-8), 128.1 (C-4a), 126.4 (C-9), 123.2 (C-7), 118.9 (C-10), 40.5 (C-1″ hidden by DMSOd6 signal), 26.4 (C-1′), 14.5 (C-2′), 13.7 (C-2″); HRMS-MS: *m*/*z* calcd. for C_16_H_16_N_2_OS (M)^•+^ 284.0983, found 284.1003 [24].

##### 2-(Benzylthio)-3-ethylbenzo[*g*]quinazolin-4(3*H*)-one (**10**) 

White amorphous powder (74%); mp 132–134 °C; ^1^H NMR (700 MHz, DMSO-*d_6_*): *δ* 8.77 (s, 1H, H-5), 8.16 (br d, *J* = 8.4 Hz, 1H, H-6), 8.15 (s, 1H, H-10), 8.07 (br d, *J* = 8.4 Hz, 1H, H-9), 7.64 (br t, *J* = 8 Hz, 1H, H-7), 7.56 (br d, 2H, *J* = 7.4 Hz, H-2′/6′), 7.54 (br t, *J* = 8 Hz, 1H, H-8), 7.35 (t-like, *J* = 7.4 Hz, 2H, H-3′,5′), 7.28 (br d, *J* = 7.4 Hz, 1H, H-4′), 4.59 (s, 2H, -CH2-7′), 4.07 (q, *J* = 7 Hz, 2H, H-1″), 1.27 (t, *J* = 7 Hz, 3H, H-2″); ^13^C NMR (175 MHz, DMSO-*d_6_*): *δ* 161.1 (C-4), 155.2 (C-2), 142.6 (C-4b), 137.2 (C-1′), 136.8 (C-9a), 130.9 (C-5a), 129.9 (C-5), 129.8 (C-3′,5′), 129.7 (C-6), 128.9 (C-8), 128.7 (C-2′,6′), 128.3 (C-4a), 127.9 (C-4′), 126.4 (C-9), 123.3 (C-7), 118.9 (C-10), 40.5 (C-1″), 35.9 (-CH2-7′), 13.8 (C-2″); HRMS-MS: *m*/*z* Calcd for C_21_H_18_N_2_OS (M)^•+^ 346.1140, found 346.1154

##### 3-(((3-Ethyl-4-oxo-3,4-dihydrobenzo[*g*]quinazolin-2-yl)thio)methyl)benzonitrile (**11**)

Pale yellow amorphous powder (81%); mp 140–142 °C; ^1^H NMR (700 MHz, DMSO-*d6*): *δ* 8.80 (s, 1H, H-5), 8.19 (br s, 2H, H-6, 10), 8.08 (br d, *J* = 8.4 Hz, 1H, H-9), 8.07 (br s, 1H, H-2′), 7.95 (br d, *J* = 7.3 Hz, 1H, H-6′), 7.74 (br d, *J* = 7.1 Hz, 1H, H-4′), 7.67 (br t, *J* = 7.3 Hz, 1H, H-7), 7.57 (br s, 2H, H-8, 5′), 4.66 (s, 2H, H-7′) 4.10 (q, *J* = 6.9 Hz, 2H, H-1″), 1.29 (t, *J* = 6.9 Hz, 3H, H-2″); ^13^C NMR (175 MHz, DMSO-*d6*; *δ* 161.1 (C-4), 155.3 (C-2), 142.6 (C-4b), 139.8 (C-1′), 136.8 (C-5a), 134.8 (C-6′), 133.4 (C-2′), 131.5 (C-4′), 130.9 (C-9a), 130.4 (C-5), 130.1 (C-5′), 129.1 (C-6), 128.4 (C-8), 128.1 (C-4a), 126.5 (C-9), 123.2 (C-7), 118.9 (C-10), 119.1 (C≡N), 111.7 (C-3′), 41.0 (C-1″), 35.6 (-CH2-7′), 13.7 (C-2″); HRMS-MS: *m*/*z* calcd. for C_22_H_17_N_3_OS (M)^•+^ 371.1092, found 371.1108.

##### 3-Ethyl-2-((3-methylbenzyl)thio)benzo[*g*]quinazolin-4(3*H*)-one (**12**)

White amorphous powder (70%); mp 186–188 °C; ^1^H NMR (700 MHz, DMSO-*d6*): *δ* 8.77 (s, 1H, H-5), 8.16 (br d, *J* = 8.3 Hz, 1H, H-6), 8.14 (s, 1H, H-10), 8.07 (br d, *J* = 8.3 Hz, 1H, H-9), 7.64 (br t, *J* = 7.6 Hz, 1H, H-7), 7.54 (br t, *J* = 7.6 Hz, 1H, H-8), 7.36 (br s, 1H, H-2′), 7.34 (br d, *J* = 7.4 Hz, 1H, H-6′), 7.23 (br t, *J* = 7.4 Hz, 1H, H-5′), 7.08 (br d, *J* = 7.4 Hz, 1H, H-4′), 4.54 (s, 2H, -CH2-7′), 4.06 (q, *J* = 7 Hz, 2H, H-1″), 2.29 (s, 3H, Ar-*Me*), 1.27 (t, *J* = 7 Hz, 3H, H-2″); ^13^C NMR (175 MHz, DMSO-*d6*): *δ* 161.1 (C-4), 155.3 (C-2), 142.6 (C-4b), 138.1 (C-1′), 136.9 (C-5a), 136.8 (C-3′), 130.9 (C-9a), 130.5 (C-5), 129.7 (C-6), 129.0 (C-2′), 128.9 (C-8), 128.6 (C-5′), 128.4 (C-4a), 128.1 (C-4′), 126.9 (C-9), 126.4 (C-6′), 123.2 (C-7), 118.9 (C-10), 41.0 (C-1″), 36.0 (-CH2-7′), 21.4 (Ar-*Me*), 13.8 (C-2″); HRMS-MS: *m*/*z* calcd. for C_22_H_20_N_2_OS (M)^•+^ 360.1296, found 360.1312.

##### 3-Ethyl-2-((3-methoxybenzyl)thio)benzo[*g*]quinazolin-4(3*H*)-one (**13**)

White amorphous powder (80%); mp 110–112 °C; ^1^H NMR (700 MHz, DMSO-*d6*): *δ* 8.80 (s, 1H, H-5), 8.19 (br d, *J* = 8.4 Hz, 1H, H-6), 8.17 (s, 1H, H-10), 8.09 (br d, *J* = 8.4 Hz, 1H, H-9), 7.66 (br t, *J* = 7.5 Hz, 1H, H-7), 7.56 (br t, *J* = 7.5 Hz, 1H, H-8), 7.27 (br t, *J* = 7.6 Hz, 1H, H-5′), 7.16 (br s, 1H, H-2′), 7.13 (br d, *J* = 7.6 Hz, 1H, H-6′), 6.85 (br d, *J* = 7.6 Hz, 1H, H-4′), 4.58 (s, 2H, -CH2-7′), 4.09 (q, *J* = 7 Hz, 2H, H-1″), 3.74 (s, 3H, -OCH_3_), 1.28 (t, *J* = 7 Hz, 3H, H-2″); ^13^C NMR (175 MHz, DMSO-*d6*): *δ* 161.1 (C-4), 159.7 (C-3′), 155.3 (C-2), 142.6 (C-4b), 138.7 (C-1′), 136.8 (C-5a), 130.9 (C-9a), 130.1 (C-5′), 129.8 (C-5), 129.1 (C-6), 128.4 (C-8), 128.1 (C-4a), 126.5 (C-9), 123.2 (C-7), 122.0 (C-6′), 118.9 (C-10), 115.5 (C-4′), 113.4 (C-2′), 55.5 (-OCH3), 41.0 (C-1″), 35.9 (-CH2-7′), 13.8 (C-2″); HRMS-MS: *m*/*z* calcd. for C_22_H_20_N_2_O_2_S (M)^•+^ 376.1245, found 376.1250.

##### 3-Ethyl-2-((2-(piperidin-1-yl)ethyl)thio)benzo[*g*]quinazolin-4(3*H*)-one (**14**) 

Pale yellow amorphous powder (50%); mp190–192 °C; ^1^H NMR (500 MHz, DMSO-*d_6_*): *δ* 8.80 (s, 1H, H-5), 8.16 (m, 2H, H-6, 10), 8.07 (br d, *J* = 7.5 Hz, 1H, H-9), 7.65 (br t, *J* = 7.5 Hz, 1H, H-7), 7.53 (br t, *J* = 7.5 Hz, 1H, H-8), 4.13 (br s, 2H, H-1″), 3.47 (br s, 2H, CH2-7′), 2.72 (br s, 2H, CH2-8′), 2.50 (m hidden by DMSO-6 signal, 4H, H-2′/6′), 1.54 (m, 4H, H-3′/5′), 1.42 (m, 2H, H-4′), 1.28 (br s, 3H, H-2″); ^13^C NMR (125 MHz, DMSO-*d_6_*): *δ* 161.1 (C-4), 155.7 (C-2), 142.7 (C-4b), 136.8 (C-5a), 130.0 (C-9a), 129.9 (C-5), 129.0 (C-6), 128.4 (C-8), 128.1 (C-4a), 126.4 (C-9), 123.1 (C-7), 118.9 (C-10), 56.5 (C-2′/6′), 54.1 (CH2-7′), 41.4 (C-1″), 31.2 (CH2-8′), 25.7 (C-3′/5′), 23.5 (C-4′), 13.7 (C-2″); HRMS-MS: *m*/*z* calcd. for C_21_H_25_N_3_OS (M)^•+^ 367.1718, found 367.1733.

##### 2-(3-((3-Ethyl-4-oxo-3,4-dihydrobenzo[*g*]quinazolin-2-yl)thio)propyl)isoindoline -1,3-dione (**15**) 

White amorphous powder (76%); mp 102–1704 °C; ^1^H NMR (700 MHz, DMSO-*d6*): *δ* 8.78 (s, 1H, H-5), 8.18 (br s, 1H, H-6), 7.96–7.80 (m, 5H, H-9, 5′/6′, 4′/7′), 7.73 (s, 1H, H-10), 7.65 (br s, 1H, H-7), 7.55 (br s, 1H, H-8), 4.10 (br s, 2H, H-1″), 3.79 (br s,2H, CH2-8′), 3.35 (br s hidden by H_2_O-signal, 2H, CH_2_-10′), 2.14 (br s, 2H, CH2-9′), 1.28 (br s, 3H, H-2″); ^13^C NMR (175 MHz, DMSO-*d6*): *δ* 168.6 (C-1′/3′), 161.1 (C-4), 155.4 (C-2), 142.6 (C-4b), 136.7 (C-5a), 134.9 (C-3a′/7a′), 132.2 (C-5′/6′), 130.9 (C-9a), 129.8 (C-5), 129.1 (C-6), 128.4 (C-8), 127.9 (C-4a), 126.4 (C-9), 123.6 (C-4′/7′), 122.9 (C-7), 118.9 (C-10), 40.5 (CH_2_-1″, hidden by DMSO-d6 signals), 37.1 (CH2-8′), 29.1 (CH_2_-10′), 28.4 (CH2-9′), 13.7 (CH_3_-2″); HRMS-MS: *m/z* calcd. for C_25_H_21_N_3_O_3_S (M)^•+^ 443.1304, found 443.1321.

Compounds **16** and **17** were reported [23].

### 3.2. Biology

#### 3.2.1. MTT Assay

The antiproliferative activity of the test compounds against HepG2, and MCF-7 human cancer-cell lines were evaluated by MTT assay. The detailed methodology procedure was reported [27,28,29]. 

#### 3.2.2. Cell-Cycle Analysis by Flow Cytometry

To investigate whether the tested compounds could arrest HepG2 and MCF-7 mitotic cell division, cell distributions at cell-cycle phases were analyzed by flow cytometry following their treatment with the vehicle or the indicated compound at its IC_50_ concentration. Cells and samples were prepared as previously reported [30], and a PI (ab139418) Flow Cytometry Kit (Abcam, Cambridge, UK) was used in accordance with the manufacturer’s protocol. In brief, both HepG2 and MCF-7 cells were separate, plated into six-well plates at a density of 0.6 × 10^6^ cells/well in DMEM growth medium supplemented with 10% FBS. Cells were incubated for 24 h at 37 °C and 5% CO_2_ prior to exposure to either the vehicle (DMSO) as the control or **13**, **14**, or **15**, at their IC_50_ concentrations. In addition, MCF-7 cells were subjected to compound **10** at 9.61 μM (its recorded IC_50_) rather than HepG2 due to its higher IC_50_ (36.2 μM) against the latter cells. All cells were incubated for 48 h, washed with cold PBS after being harvested to single-cell suspensions by trypsinization, and fixed with 70% ethanol. The fixed cells were rehydrated in PBS prior to being lysed in a prepared solution of PI, Triton X-100, and RNase. Finally, the cells were kept at 37 °C for 15 min and analyzed using a FACS Caliber flow cytometer (BD Biosciences and Co., Franklin Lakes, NJ, USA).

#### 3.2.3. Measurements of Apoptosis Using Double Staining with Annexin-V FITC/PI Dyes

Apoptosis was detected for both HepG2 and MCF-7cells using the same preparation and sample treatment conditions as those used for cell-cycle analysis. The manufacturer’s instructions for the annexin-V FITC apoptosis detection kit (K101, Biovision, Milpitas, CA, USA) were followed. In brief, treated cells were detached after 48 h by trypsinization, washed twice with cold PBS, and stained with 5 μL of annexin-V fluorescein isothiocyanate (annexin-V FITC) and 5 μL of PI in the binding buffer for 15 min at 37 °C. Finally, the processed cells were analyzed using a FACS Caliber flow cytometer (BD Biosciences and Co., Franklin Lakes, NJ, USA).

#### 3.2.4. Hoechst 33258 Nuclear Staining-DNA Fragmentation

In the case of detecting the induction of apoptosis in HepG2 and MCF7 cells by the fluorescent microscopy, also same conditions of preparations and sample treatment as in the section of cell-cycle analysis were followed, while the instructions of Hoechst 33258 Staining Dye Solution, Cat. No. #ab228550 (Abcam, Cambridge, UK) were performed. Fluorescent micrographs were taken by a Fluorescence Microscope (LF-302) Cat. No. #2917 at 40× objective lens.

#### 3.2.5. In Vitro Inhibition of VEGFR-2

Angiogenesis inhibition was investigated using an enzymatic VEGFR-2 (KDR) Kinase Assay Kit Cat. #40325 assay following the manufacturer’s instructions [31]. A series of 10-fold dilutions were prepared from compounds **10**, **13**–**15** to provide the concentration range 0.0–10.0 μM. Sorafenib (Cat No. 284461-73-0, Santa Cruz, Dallas, TE, USA) was used as a standard inhibitor to the VEGFR2 enzyme. At the indicated end of the experiment, the luminescence was measured by a Tecan spark microplate reader. A dose-response curve was constructed for every tested compound, and the IC_50_ value, i.e., the concentration that inhibits VEGFR-2 enzyme activity to 50%, was also determined using Prism v.6 curve fitting software (GraphPad Software, Inc., San Diego, CA, USA).

### 3.3. Docking Study

MOE software was used in this study [5]. The 4ASE [8] (sorafenib) crystal structures of VEGFR-2 were used to study the binding modes between the docked-selected compounds and amino acid residues in the active site of VEGFR-2. The protein structure of VEGFR-2 was obtained from the Protein Data Bank [PDB Code 4ASE].). The structures of the synthesized and reference compounds were constructed using MOE-Builder. The related 3D structures were also obtained, and the energies of the identified molecules were minimized using the default parameters of MOE energy minimization algorithm.

A maximum of 10 conformations were allowed for each ligand using the default parameters of MOE (placement, triangle matcher; rescoring 1, London dG; refinement, force field; rescoring 2, GBVI/WSA dG). The 300 top-ranked conformations of the docked compounds were saved in a separate database. 

### 3.4. QSAR Methodology

From several published reports and the chembl-database website, 110 and 90 known MCF-7 and HepG2, respectively, compounds were collected and their activities described in terms of inhibitory activity (pIC_50_ nM) [32,33,34,35]. The 17 benzoquinazoluine compounds and the reference compound were imported as .mdb files into a molecular database in MOE Version 2015.1001) including molecular structures and activity values.

PLS regression was employed as a statistical method to evaluate structure-activity relationships in QSAR. The QSAR models were built and cross-validation was conducted to check the reasonable QSAR models. Various statistical parameters (Appendix A) were utilized for validation of model suitability for the prediction of the anti-proliferative activities of the target compounds. This includes the correlation coefficient (*R*^2^), which describes the fraction of the total variation attributed to the model, and the cross-validation coefficient parameter ($Q_{\mathrm{CV}}^{2}$).

## 4. Conclusions

In conclusion, new benzo[*g*]quinazolines (**1**–**17**) were synthesized and evaluated for their in vitro antiproliferative activity against two selected human cancer cell lines, namely HepG2 and MCF-7. Most of the benzoquinazolines demonstrated promising activity. Compounds **13**–**15** showed the highest activity against HepG2 and MCF-7, and their effects were further investigated by cell-cycle analysis, revealing a similarity in the activity characteristics of **13** and **14** against both MCF-7 and HepG2, involving the targeting of the G1 and S phases, respectively. The best induction of apoptosis in MCF-7 cells was observed for compound **14**. Molecular docking confirmed that 13 and 15 are type-II VEGFR-2 inhibitors and experimentally, gave 1.5- and 1.4-fold inhibition relative to the standard sorafenib. In QSAR analysis, the predicted pIC_50_ values of the active compounds showed a significant correlation with the experimental values from principal component analysis plots. These results strongly demonstrate the anticancer activity of benzoquinazolines **10**, **13**–**15** in vitro.

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
