# Peer review of "Antiproliferative and Antiangiogenic Properties of New VEGFR-2-targeting 2-thioxobenzo[g]quinazoline Derivatives (In Vitro)"

_molecules, 2020, doi:10.3390/molecules25245944_

Round 1

Reviewer 1 Report

Although the synthesis has been already published and the IC50 values are not much better than the control molecule, the Cell-cycle analysis by flow cytometry, the Measurements of apoptosis and the Docking study provide a very valuable information for future studies.

The paper in general is well written, nevertheless it is necessary a careful revision of the spelling all along the manuscript, for example “autophosphor ylation” in paragraph 64 or “benzo[f] quinazoline” in paragraph 77. Furthermore according with the IUPAC naming rules it should not be a hyphen in thioxo-benzo[g]quinazoline, it should be thioxobenzo[g]quinazoline, it should be corrected in the Title and in the rest of the article.

The description of the results should be in past tense, for example it should be “presented” in paragraph 141 or exhibited in paragraph 143.

Author Response

Dear Editor of Molecules

Thank you so much for your fine cooperation. According to the reviewers comments, we have carefully revised and corrected our manuscript. All corrections were highlighted with yellow color throughout the manuscript point by point as following:

 Reviewer 1

Although the synthesis has been already published and the IC50 values are not much better than the control molecule, the Cell-cycle analysis by flow cytometry, the Measurements of apoptosis and the Docking study provide a very valuable information for future studies.

The paper in general is well written, nevertheless it is necessary a careful revision of the spelling all along the manuscript, for example “autophosphor ylation” in paragraph 64 or “benzo[f] quinazoline” in paragraph 77. Furthermore according with the IUPAC naming rules it should not be a hyphen in thioxo-benzo[g]quinazoline, it should be thioxobenzo[g]quinazoline, it should be corrected in the Title and in the rest of the article.

Response:  We have corrected and all corrected names were indicated by yellow colour.

The description of the results should be in past tense, for example it should be “presented” in paragraph 141 or exhibited in paragraph 143.

Response: the whole paragraph was revised and corrected in past tense.

Date of this response

12 Dec 2020

Reviewer 2 Report

This paper reports the antiproliferative, antiangiogenic and antiapoptotic activities of seventeen componds, based on a benzoquinazoline structure, against two human cancer cell lines: breast cancer cell line MCF-7 and liver cancer cell line HepG2. The compounds were readily synthesised from 3-amino-2-naphthoic acid in one or two steps. This work is an extension of the work presented in Ref 13 (DOI: 10.3987/COM-15-13282), where the cytotoxicities of similar derivatives were investigated.

In initial MTT assays, compounds 13, 14 and 15 were the most promising against HepG2, whilst 11, 13, 14, 15 were the most promising against MCF-7. These compounds were investigated futher. Compound 14 was found to be best inducer of apoptosis in MCF-7, and 13 and 15 were found to be the best VEGFR-2 inhibitors (VEGFR-2 inhibition is known to prevent angiogensis). The experimental results correlate with a computational QSAR study.

I would recommend this manuscript for publication in Molecules if the following issues are addressed:

  1. Title: I think the reader would benefit from 'in vitro' added to the end the title 
  2. Figures 1 and Scheme 1: The ChemDraw figures should be improved, and be of consistent size and font.
  3. Scheme 1: The compound labelling should be improved. From just the scheme, it is not clear that 1 and 2 are the top-centre compound, 16 and 17 are the top-right compound, and 3-15 are the bottom-centre compound. I would suggest moving the compound number keys closer to the relevant compounds, so that the numbering is explicitly clear to the reader.
  4. It would benefit the reader if the nature of HepG2 and MCF-7 cells was stated, i.e. liver and breast cancer cell lines, in the introduction
  5. Section 2.1 Line 110: '3-aminonaphthaoic' should be changed to '3-amino-2-naphthoic acid', to remove the structure ambiguity in the name.
  6. Section 2.1 Line 112: Change 'alkyl(heteroalkyl)halide' to 'alkyl / heteroalkyl halide'
  7. It would also be informative to carry out the viability assays on non-cancer human cell lines, to see if there is selectivity, or if the compounds are just generally toxic to cells in general, cancer cells and non-cancer cells.
  8. Page 23, line 537: Finish the final sentence with 'in vitro'.
  9. Section 3.1.1: As the authors state, the procedures are reported previously. However, the compounds presented here are previously unreported. Therefore, I would prefer to see the general procedures repeated in this section.
  10. The page numbering should be corrected.
  11. The manuscript would benefit from an abbreviations list.

Author Response

Dear Editor of Molecules

Thank you so much for your fine cooperation. According to the reviewers comments, we have carefully revised and corrected our manuscript. All corrections were highlighted with yellow color throughout the manuscript point by point as following:

Reviewer 2

  1. Title: I think the reader would benefit from 'in vitro' added to the end the title 

Respnse: it was added to the title and highlighted with yellow color

  1. Figures 1 and Scheme 1: The ChemDraw figures should be improved, and be of consistent size and font.

Response: It was done and fixed

  1. Scheme 1: The compound labelling should be improved. From just the scheme, it is not clear that 1 and 2 are the top-centre compound, 16 and 17 are the top-right compound, and 3-15 are the bottom-centre compound. I would suggest moving the compound number keys closer to the relevant compounds, so that the numbering is explicitly clear to the reader.

Response: It was amanded

  1. It would benefit the reader if the nature of HepG2 and MCF-7 cells was stated, i.e. liver and breast cancer cell lines, in the introduction

Response: we have added the liver and breast in abstract and introduction  and highlighted with yellow color

  1. Section 2.1 Line 110: '3-aminonaphthaoic' should be changed to '3-amino-2-naphthoic acid', to remove the structure ambiguity in the name.

Response: It was corrected and highlighted with yellow color

  1. Section 2.1 Line 112: Change 'alkyl(heteroalkyl)halide' to 'alkyl / heteroalkyl halide'

Response: it was corrected and highlighted with yellow color

  1. It would also be informative to carry out the viability assays on non-cancer human cell lines, to see if there is selectivity, or if the compounds are just generally toxic to cells in general, cancer cells and non-cancer cells.

Response: The data presented in this manuscript was preliminary results.  Detailed experiments are being carried out in order to investigate further modes of action and the mechanisms by which these compounds exert their effect as well as the safety of these compounds will be studies as well on at least one human healthy cell line. These data, when finalized, will be published elsewhere.

  1. Page 23, line 537: Finish the final sentence with 'in vitro'.

Response: it was added and highlighted with yellow color

  1. Section 3.1.1: As the authors state, the procedures are reported previously. However, the compounds presented here are previously unreported. Therefore, I would prefer to see the general procedures repeated in this section.

Response: It was done.

  1. The page numbering should be corrected and 11-The manuscript would benefit from an abbreviations list.

Response: the abbreviations  are already know.

Date of this review

11 Dec 2020 18:42:04

Best regards

Alsalahi Rashad/ corresponding author